# Audiovisual Competences in Times of COVID-19: The Role of Educational Actors in Media and Digital Learning of Adolescents

Abel Suing [1,*], Juan-Pablo Arrobo-Agila [1], Ximena Coronado-Otavalo [2], Viviana Galarza-Ligña [2] and Amparo Reascos-Trujillo [2]

1  Departamento de Ciencias de la Comunicación, Facultad de Ciencias Sociales, Educación y Humanidades, Universidad Técnica Particular de Loja, Loja 110107, Ecuador
2  Escuela de Ciencias Sociales y Humanas, Pontificia Universidad Católica del Ecuador-Ibarra, Ibarra 100112, Ecuador
*  Correspondence: arsuing@utpl.edu.ec

**Abstract:** This research arises from the need to know the elements that have an impact on the audiovisual competencies of adolescents during the confinements provoked to prevent the spread of COVID-19. The purpose is to diagnose the development of audiovisual communication skills among Ecuadorian adolescents as a contribution to sustainability, based on the intervention of educational actors. The methodology is qualitative, with a descriptive approach. The instruments used were: focus groups with parents, students, and teachers from public and private institutions in Ecuador; semi-structured interviews with experts: and non-participant observation. It can be concluded that adolescents acquire audiovisual skills, processes, and languages autonomously before entering college, but they do so without the social context, ethical values and responsibilities that should be part of complete media learning. The demands of online learning during the pandemic have resulted in the development of skills and attitudes, but they have not led to full media literacy. It is pertinent to provoke innovations and updates in the curricula of higher education, specifically in the careers of social communication, in order to be aware of the technological changes on the basis of deontological principles and in favour of democratic values, tolerance, and responsibility with the sustainability of nature and people.

**Keywords:** audiovisual and media competences; COVID-19; digital education; educational convergence; ubiquitous learning

## 1. Introduction

Since the COVID-19 pandemic began, many adolescents have had to develop their skills and competencies in order to participate in virtual and distance learning modalities. The physical distance of classrooms and lack of personal contact have led to new perspectives and demands on higher education based on the audiovisual domains that high school students have demonstrated.

In the context of the pandemic, due to COVID-19, the educational community was faced with the intense challenge of digitalization and virtualization of school activities. In the midst of the crisis, issues related to media and information competences took centre stage in the lives of teachers, students, and families [1].

Recent studies on young people's media skills conclude that "they have an important development that appeals to the dimension of critical understanding and thus to autonomy, which is made explicit in their ability to be critical beyond technology, the media and social networks" [2] (p. 109). Based on this, "they have a significant development that appeals to the dimension of critical understanding and thus to autonomy, which is made explicit in their ability to be critical beyond technology, the media and social networks" [2] (p. 109), from which we can see the urgency of closing the gap between the capacities of teachers

and learners, since students "point out that while in their daily lives they take photos, record videos, edit, etc., in their studies they are limited to do what their teachers ask them to do" [3] (p. 124).

The ease of communication through the Internet, the increase in distance and virtual learning, and the growing offer of online entertainment have led many children and adolescents to acquire skills autonomously in the management of information, without adequate control or systematization [4]. This situation has worsened since the onset of the COVID-19 pandemic, deepening the socio-cognitive deficiencies in the understanding of media narratives and convergences [5].

Both traditional and modern online media have a significant impact on audiences, and as a result, educational systems seek to develop critical consumption skills in individuals. However, challenges persist in terms of lack of understanding and appropriate use of media content.

At the beginning of the 21st century, education experts anticipated that "the power of communication" [6] (p. 117) should be taught as early as possible, i.e., to develop a media education that enables competences "to use and interpret the media" [7] (p. 71).

The need for media skills was anticipated in the Delors Report [8], which stated that the education of the future would have "a twofold requirement, that of transmitting, massively and effectively, an increasing amount of evolving theoretical and technical knowledge, and that of defining orientations". In this context, it should be mentioned that Article 19 of the Universal Declaration of Human Rights states that everyone has "the right to freedom of expression; this right includes freedom to hold opinions without interference and to seek, receive and impart information and ideas through any media and regardless of frontiers"; therefore, media and information literacy (MIL) provides citizens with the skills they need to seek and enjoy the full benefits of this fundamental human right.

Worldwide, many organizations use the term media education, which is sometimes accepted to mean both media literacy and information literacy. UNESCO uses the term MIL in order to harmonize the different notions [9].

Media literacy, adopted by the European Commission in 2007, is defined as the ability to "read, analyse and evaluate the power of the images, sounds, and messages we are currently confronted with in our daily activity [...] as well as the ability to communicate competently through the media available to us" [10] (p. 42).

The educational importance of traditional media was highlighted in the Grunwald Declaration in 1982, where political and educational systems are called upon to assume their obligations to promote a critical understanding of communication phenomena among citizens [11]; it is furthermore pointed out that school and family have a co-responsibility to prepare young people for a world dominated by audiovisual images. The integration of media literacy into education systems is an important measure of educational quality.

Media and information literacy is identified as an educational requirement by international organizations such as UNESCO and the European Union. In the report "Media and information literacy for all and by all", the need to increase the basic skills for communicating and understanding the ideas that circulate in technological environments was emphasized [12].

The European Commission on "media literacy" defined the term Media Competence as "the ability to access media, understand and critically evaluate different aspects of media and multimedia content in order to create communication in a variety of contexts" [13] (p. 2).

Although there are differences in the skills required, the general term for all of them is communicative or media competence, which refers to "the capacity of an individual to interpret and analyse images and audiovisual messages from a critical reflection, and to express him/herself with a minimum of correction in the communicative field" [14] (p. 10), while audiovisual competences are subsumed in the definition of media competences [15].

According to Ferrés and Piscitelli [16] (p. 78), media competencies refer to the set of knowledge, skills, abilities and attitudes that people use when interacting with the media

"critically with messages produced by others, being able to produce and disseminate their own messages". In this sense, six dimensions are proposed, taking media literacy as a starting point: languages, technology, production and dissemination processes, reception and interaction processes, ideology and values, and the aesthetic dimension, approaching its content from the areas of expression or production of messages and reception or understanding of media content [14,16,17].

In relation to the media, competences must contribute to the development of citizens' personal autonomy, as well as their social and cultural commitment [18]. Therefore, it is understood that media literacy should "necessarily address key issues of our time, such as the phenomena that affect personal autonomy and decision-making capacity, and promote the formation of a critical awareness of the new scenarios created by the media" [19] (p. 48).

Educational institutions, especially schools and universities, have a fundamental role to play in the development of media literacy [20], because they promote critical thinking, creativity and citizenship awareness [21]. Schools and universities "play a central role as disseminations of media education by having to adapt [...] the skills necessary for students to function in this media scenario" [22] (p. 89).

Communication professionals, and more specifically, audiovisual communication professionals, have seen how in the last decade, digitalization and the consequent appearance of new formats, new roles for audiences and new ways of distributing content have led to profound changes in production structures and ways of working [23]. Additionally, "it seems that the curricula of university degrees in audiovisual communication are not sufficiently adapted to this new context" [24] (p. 1537). It is necessary to make a firm commitment to the conquest of a new competence that allows greater access to basic rights for all citizens [25].

Secondary education may be facing a lack of commitment from academic leaders. However, it is crucial that innovative mechanisms, tools and resources are adopted immediately to drive dynamic and up-to-date learning [26]; however, "the impact of digitalization has been particularly decisive in the professional field [...] whose current academic orientation requires a review of subjects, content and, above all, of the competences committed to in the current curricula" [27] (p. 370).

Already in the 20th century, the rise of audiovisual language raised the need for audiovisual and media literacy [28]. This new literacy struggled for a place in compulsory education. According to Buckingham, "there were times when it seemed as if media education was about to become a fundamental right for all young people. But it never happened, at least until now" [29].

It is relevant to point out that in the era of post-truth, big data and artificial intelligence, media education is necessary, but not sufficient, to combat disinformation and manipulation [30,31]. The COVID-19 pandemic brought with it an increased media presence and consequent infoxication, which reopens the debate on the obligatory nature of media education and new parental roles [32].

In an increasingly complex and constantly changing world, it is essential that training in media and audiovisual skills has a vision of the future that offers a humanistic education. This will enable citizens to acquire the competencies and skills necessary to function effectively in their social and professional environment, thus contributing to the development of a more informed, critical and participatory society [33]. The aim is to achieve "a critical and reflective attitude, based on the participation and interrelation of citizens" [34] (p. 38).

On the basis of the above, the relevance of carrying out a research that identifies the audiovisual competences that adolescents formed autonomously in order to continue participating in the teaching and learning processes during the first years of control of the spread of COVID-19 is justified; in addition, we seek to know the perceptions of parents and teachers related to the formation and application of the digital audiovisual competences of their children and students, with the intentions of understanding and anticipating the scenarios where higher education should be implemented.

The case of Ecuador is investigated as a case study, because it will allow important conclusions to be drawn that will have immediate application in the educational communities of Andean and Latin American countries, as they share a similar language and culture and similar educational contours.

## 2. Literature Reviewing

Competency formation is related to connectivism, since "at its heart, connectivism is the thesis that knowledge is distributed across a network of connections, and therefore that learning consists of the ability to construct and traverse those networks" [35] (p. 85), and skills developed empirically by adolescents or in turn guided by web-based resources.

This approach leaves aside the traditional position because "constructivism assumes that learners are not simply empty vessels to be filled with knowledge" [36] (p. 3). Education needs a reinvention based on the changes brought about by the digital revolution and new learning ecosystems.

The move towards the knowledge society means that learning is acquired in different spaces and realities, imperatively aided by the use of technological devices with access to the digital world. A constant in today's learning, according to Sakamura and Koshizuka, is that "we learn about anything at anytime, anywhere using ubiquitous computing technology and infrastructure. Thus, essential subject for learning exist in our daily living environment, but not in classrooms or textbooks" [37] (p. 13).

In such a way that traditional spaces are not the only places where young people develop competences, learning today breaks the limits of the school and moves to other environments. However, when information is ubiquitous and omnipresent, it is appropriate to take a look at invisible learning, which questions the direction of education towards what the world expects of it. Education "demands an ecological, systemic, long-term and inclusive improvement" [38] (p. 20), and bridges are needed between school and digital metaverses. If it is accepted that "invisible learning is an open and provocative dialogue, which seeks to rethink the temporal and spatial limits that have been adopted so far to understand education" [20] (p. 25), then it is possible to link the "innovative and technological" vision of adolescents in new learning, in times of uncertainty [39].

According to the above, it is urgent and necessary to try to incorporate debates and good practices of audiovisual competences from higher education institutions due to the implicit role they have in sustainable development, as demonstrated in previous research [40–43]. Universities play a fundamental role in promoting the principles of sustainability, contributing to the paradigm shift towards a more sustainable present and future; they are committed to designing institutional policies and appropriate strategies that facilitate the "change of the educational and academic model that allows for genuine inter- and transdisciplinary training, contemplating the development of competences specific to a profession, general competences and the competences demanded by sustainable development" [44] (p. 269).

Many interests have influenced a misunderstood education to deliver instructions to generate economic growth, which is why it is necessary to implement policies and incentives that lead to maintaining and improving educational models relevant to human and sustained progress: because "the current model of progress is based on outdated narratives and false myths that teach unsustainable models for life on the planet because they perpetuate inequality and domination" [45] (p. 155).

To manage education "towards the recognition of human and social complexity, as well as to promote sustainability, is to open the possibility of building a better world, lasting in time [...] allowing the passage towards a more just society" [46] (p. 1499); in this intention, the participation of committed teachers is essential to confront intolerable social realities "to reactivate citizenship, to think of another different world and to walk towards it. In other words, it is not simply a matter of training good architects, but better human beings. To train the best people for and by the world" [47] (p. 42).

At the macro level, "educational policies must be thought through with a view to the future, in order to take on the challenge of moving towards sustainability. In this process, the academy, understood as schools, high schools, universities, will set their plans, goals and specific objectives, but they must have social relevance" [46] (p. 1499).

Achard [48] states that "it is not a matter of continuing to give classes, including digital technological resources; it is necessary to generate substantive modifications in teaching practices, and in the quantity and quality of learning (more and better), enhancing student autonomy" (p. 51). In this sense, authors such as Esteve, Castañeda and Adell [49] stated that the digital competence of the teacher goes beyond the generic handling of platforms; it refers to a deep knowledge of the digital, its demands and complexities, to take advantage of it in their training mission; therefore, the teacher finds on the internet an ally that allows them to update themselves in innovation [50] and to know the tools and models offered by the virtual universe [51].

As a result of the health emergency caused by the COVID-19 pandemic, schools adjusted curricula to respond to the imminent changes, and the Ministry of Education of Ecuador issued guidelines in which it stated that digital competences education focuses on the construction of digital citizenship. The management of technologies is not only transversal to all areas of knowledge, but also from new digital skills [to] train people with creative initiatives, capable of solving problems collaboratively [52].

It should also be noted that "digital competences encompass computational thinking, which is understood as the process by which an individual, through critical thinking, knows how to identify a problem, define it and find a solution" [53] (p. 8). This is coupled with the urgency of a shift in education systems towards the attainment of hard skills and soft competences.

Competency-based education focuses on developing skills such as creative problem solving, the ability to search for and select relevant information, the effective execution of tasks, critical analysis and reflection in response to the demands of the environment. This approach allows students to apply the knowledge acquired in real situations and prepares them to face the challenges of today's world [54,55]. "Each competence, apart from cognitive knowledge and skills, also includes skills and attitudes, willingness and ability to learn" [56] (p. 455). The concept of competence "was first associated with the world of work, then gradually became related to the academic world. Competence is generally understood as a combination of knowledge, skills, and attitudes that are considered necessary for a given context" [16].

In short, in this contextualization, media education is understood as the advancement of a critical attitude and participation, qualities that favour creativity, especially in young people [7]. Through it, the subject possesses attributes that enable them to value and interpret the enormous number of forms of communication. Buckingham [57] proposes four major dimensions for media literacy: production, representation, audiences and language; "a much broader concept of media literacy, based on critical thinking about the economic, ideological and cultural dimensions of media, needs to be handled" [58] (p. 213).

Students in the current era must acquire skills to cope in a digital environment saturated with information, in which they must be able to analyze and make decisions in a diverse society, characterized by a multiplicity of screens, media, languages and emerging technologies [57]. Likewise, "exercising full, responsible and critical citizenship today implies that people must be more and better prepared to interact with a complex, diverse and interrelated context such as that of this new millennium" [18] (p. 115).

## 3. Materials and Methods

### 3.1. Objective

The purpose of the research is to diagnose the development of the audiovisual communication skills of Ecuadorian adolescents during the COVID-19 pandemic, as a contribution to sustainability, based on the intervention of educational actors.

The research question is: Do the audiovisual competencies acquired in self-taught learning by Ecuadorian adolescents during the confinements that took place to reduce the spread of COVID-19 constitute an integral capacity to perform in society and involve changes in the curricula of higher education, specifically in social communication courses?

### 3.2. Methodology

The research methodology is qualitative and descriptive, and it focuses on analysing a situation at a given moment in time, in this case, media competence. It also describes variables and assesses their incidence [59]. This approach is considered ideal for evaluative research [60]. It should be noted that when attempting to understand a social phenomenon such as educational events, it is appropriate to analyse, observe and ask questions [61].

The researching instruments used are focus groups, semi-structured interviews and non-participant observation. It was considered relevant to use these qualitative research instruments because they addressed an emerging phenomenon that had not occurred before, involving the subjectivity of the people addressed. On the basis of respect, the guidelines of the Ethics Committee for Research on Human Beings, CEISH-UTPL, and the accompaniment of parents and teachers, information generated by adolescents was collected in safe environments and under supervision, both at school and in the MediaLab laboratory, where non-participant observation was carried out.

### 3.2.1. Focus Group Discussions

The researchers complied with the "informed consent" protocol, signed by the parents, to gain access to the adolescents in the focus groups. Contact with the participants in the focus groups was made by means of e-mail requests, following authorization from the heads of the educational institutions. The selection of educational institutions was made on the basis of the variable "ownership" of the educational statistics of Ecuador.

Nine focus group discussions (GDD), each lasting one hour, were conducted with adolescents, parents and teachers. Residents of the provinces of Loja, Imbabura and Santo Domingo de los Tsáchilas were selected because they are representative sites of cultural diversity, located in different cardinal points, and because they are close to the universities that sponsored the study. The participants did not know each other, nor did they have a close relationship with the moderator.

The selection criteria for each GDD category were: (1) parents of adolescents (12 to 18 years old), whose children study in public or public schools; (2) adolescents between 12 and 18 years old, and students from public and public schools; and (3) teachers of Higher General Basic Education or Unified General Baccalaureate, from public, public-commissioned or public schools.

The identification codes are:

E-L-# (Student Loja—participant number)
E-Ib-# (Student Ibarra—participant number)
E-SD-# (Student Santo Domingo—participant number)
Pd-L-# (Parent—Loja—participant number)
Pd-Ib-# (Padre Ibarra—participant number)
Pd-SD-# (Parent—Santo Domingo—participant number)
P-L-# (Teacher—Loja—participant number)
P-Ib-# (Teacher—Ibarra—number of participant)
P-SD-# (Teacher—Santo Domingo—participant number)

The distribution of participants corresponds to: (a) 25 students: 11 males and 14 females; 9 live in Loja, 7 in Ibarra and 9 in Santo Domingo; average age: 25 years; (b) 17 parents: 3 males and 14 females; 6 live in Loja, 6 in Ibarra and 5 in Santo Domingo; average age: 41 years; and (c) 24 teachers: 13 males and 11 females; 10 live in Loja, 10 in Ibarra and 4 in Santo Domingo; average age: 39 years. The 66 participants in the three GDD classified according to city of residence are located as follows: 25 in Loja, 23 in Ibarra and

18 in Santo Domingo. The type of ownership of the educational establishments related to the participants is: 19 public, 6 fiscal-commissioned schools and 41 private schools.

From a theoretical perspective, a focus group is a practice and interactive strategy of social research [62,63], "consisting of bringing together a group of six to ten people and provoking a discussion among them on the topic of interest, which must be led by a moderator" [64] (p. 150); it is "ideal for capturing dominant representations, values, affective and imaginary formations, it allows for the reconstruction of social meaning, it reproduces a certain macro-social situation on a given scale" [65]. It allows the expression of different positions and attitudes of the participants, the exchange of information and the orientation of the discourse on the reality to be investigated [66].

Online focus groups were conducted because of the mobility restrictions that remain in order to reduce the spread of COVID-19, "conducting online focus groups is logistically feasible. Social researchers currently have a range of technological and communicative resources available to us that we can manage and configure to shape group dynamics" [67] (p. 112).

The topics covered in the GDD come from the research "Media Competence. Research on the degree of competence of citizens in Spain", directed by Joan Ferrés i Prats, professor at the Universitat Pompeu Fabra (Barcelona).

However, the GDD should be complemented by other techniques, such as interviews and observation [68]. Interviews are recommended to obtain direct information from key persons, and when it is desired to inquire about a subjective personal experience [65]. Interviews "will allow for a qualitative and nuanced expression of the information obtained, while serving as a contrast, confirmation, and triangulation of information [...] on those dimensions of media competence that are more complex" [69] (p. 128).

The methodological approach and instruments are also justified on the basis of a recent experience in Ecuador, where "media competencies in parents were determined. The technique of focus groups was used in order to delve deeper into the description of media competencies [...] in which five to ten people participated" [70] (p. 17).

The questions posed to parents were: (1) Do you know how to take photographs, videos or audio recordings? (2) How do you live with your children in terms of audiovisual consumption and creation? (3) Did you learn audiovisual production to help your children? (4) What audiovisual competences would you like your children to strengthen? (5) What audiovisual competences would you like your children to strengthen?

The questions addressed by the adolescents are: (1) Can you make podcasts, videos for YouTube or capture images? (2) Do you plan to capture photographs or videos? (3) What routine do you follow to capture videos or photographs? How did you get started with audiovisual technologies? (4) How do you overcome technical problems, if any, when filming? (5) Images are projected with logos of audio and image editing software: which ones have you used? (6) Which software are you interested in learning? (7) Did the use of audiovisual editing software increase while receiving virtual classes? (8) Which skill related to audiovisual production do you want to strengthen?

The themes that guided the teachers' GDD are: (1) Did you ask for assignments in audiovisual format? (2) Do you estimate that your students have acquired photo and video editing skills, since the beginning of the pandemic? (3) How did you guide your students to develop audiovisual assignments? (4) What audiovisual competences do you expect your students to strengthen at university?

### 3.2.2. Semi-Structured Interviews

In order to go deeper into the subject and obtain a wide complete understanding, as well as a personal vision of the object of the research, it was decided to apply the instrument of semi-structured interviews. The semi-structured interviews were conducted with experts whose profiles are shown in Table 1, corresponding to communication teachers who teach at Andean universities. The dialogues were carried out individually, and not in groups,

due to their backgrounds, professional profiles and activities at different times, which prevented them from coinciding, even virtually.

**Table 1.** Profiles of interviewees.

| Code | Country | Profiles |
|---|---|---|
| Interviewee-1 | Ecuador | Professor at the Universidad Politécnica Salesiana (Ecuador). PhD in Communication. Member of the university laboratory (GAMELAB-UPS). Member of the Research Group on Missions and Indigenous Peoples (GIMPI). |
| Interviewee-2 | Bolivia | Professor at the Universidad Mayor de San Andrés. Social communicator. Master's degree in communication and development. Studied at the Universidad Católica Boliviana San Pablo (Bolivia), UNED (Spain) and Universidad Andina Simón Bolívar (Bolivia). |
| Interviewee-3 | Perú | Research professor at the Universidad Privada del Norte (Peru). Doctor in Communication, professional journalist. Member of the National Anti-Corruption Network—Lambayeque. Free expression monitor at the Instituto Prensa y Sociedad de Perú and methodologist at IFEX—ALC (2011–2014). |
| Interviewee-4 | Perú | Lecturer at the Universidad Nacional del Santa (Peru). Consultant in audiovisual marketing and brand management. Associate Director of the FX MEDIA Communications Agency. Member of the Ibero-American Network of Audiovisual Narratives and the StoryCode Peru Network. |
| Interviewee-5 | Colombia | Sociologist from the Universidad Popular del Cesar (Colombia). Master's degree in visual anthropology and anthropological documentary from FLACSO (Ecuador) and PhD in social and cultural anthropology from the Universidad Autónoma de Barcelona. |
| Interviewee-6 | Ecuador | Audiovisual producer. Senior lecturer at the Universidad Técnica Particular de Loja. Doctor in Communication and Journalism at the University of Santiago de Compostela- Spain. |
| Interviewee-7 | Perú | PhD in Communication from the Pompeu Fabra University in Barcelona. Master's degree in Communication and Education from the Autonomous University of Barcelona. Degree in Communication from the University of Lima, where he is an associate professor. Project advisor at Fundación Telefónica. |
| Interviewee-8 | Bolivia | Degree in Social Communication Sciences from the San Francisco Xavier University of Chuquisaca. Master's degree in communication and educational technologies from the Latin American Institute of Educational Communication (ILCE) and the Simón Bolívar Andean University (Bolivia). |
| Interviewee-9 | Colombia | PhD in Communication from the University of Huelva. Social communicator. Specialist in organisational communication management. Member of the editorial boards of the journals Diálogos (FELAFACS) and Comunicar. Member of the International Network of Communication Historiographers. |
| Interviewee-10 | Ecuador | Master's degree in teaching and educational evaluation. Doctor in Educational Sciences. Lecturer in the Department of Educational Sciences at the Universidad Técnica Particular de Loja and Coordinator of the Master's Degree in Pedagogy. |

The semi-structured interviews were conducted with experts, whose profiles are shown in Table 1, corresponding to communication teachers who teach at Andean universities. The interviews were conducted between April and May 2022 via video calls (Zoom, Google). The questions are: (1) Will the young people, who studied during the confinements caused by COVID-19, demand new knowledge of audiovisual language at university? (2) Should the academic offerings related to audiovisual be updated to cater to a new profile of young people? (3) What areas should be strengthened in communication careers to train audiovisual competences? (4) Should university education in communication focus on deontology, without considering instrumental competences?

3.2.3. Semi-Structured Interviews

Finally, the last instrument used in this research is non-participant observation, which took place between 28 March and 1 April 2022, for which a workshop was organized. The purpose of the workshop was to construct a video based on the intervention of the adolescents. Evaluation rubrics were applied during and after the workshop. The place of observation is the MediaLab laboratory of the Universidad Técnica Particular de Loja. Working hours: 16:00 to 17:30. The first two days were given contents on audiovisual script,

planning and production, and equipment handling; the third day was left to form the work team; and between the fourth and fifth day, the script was created, images were captured and edited and the audiovisual production was presented. The profiles of the adolescents are as follows:

- Participant 1: male, attends 10th year of General Basic Education at the Amauta School.
- Participant 2: male, attends the 10th year of General Basic Education at the Cordillera School.
- Participant 3: female, attends the 9th year of General Basic Education at the Mariana de Jesús School.
- Participant 4: female, attends the 10th year of General Basic Education at the Amauta School.
- Participant 5: male, attends the 10th year of General Basic Education at the Amauta School.
- Participant 6: male, attends the 1st year of the Bachillerato General Unificado at the Santiago Fernández García School.
- Participant 7: female, attends the first year of the Unified General Baccalaureate at the Beatriz Cueva School.
- Participant 8: female, attends the 2nd year of the Unified General Baccalaureate at the Bernardo Valdivieso School.
- Participant 9: male, attends the 10th year of General Basic Education at the Daniel Álvarez School.
- Participant 10: female, attends the 2nd year of the Unified General Baccalaureate at the Bernardo Valdivieso School.

In education, the workshop is considered to be "an appropriate methodology to achieve training objectives on specific topics, based on the participants' prior knowledge, collective discussion, and the integration of theory and practice" [71]. In this case, the aim was to integrate a coherent process to create audiovisuals following an order (pre-production, production and post-production). This methodological tool makes it possible to link the knowledge of each session taught [72].

## 4. Results

The testimonies and data obtained through GDD, interviews and non-participant observation were processed manually by the research team. Magnetic and digital records were transcribed, and then common categories were located to establish relationships and identify the arguments with which to answer the research question. There were also redundancies and material that was not subject to planning that was left out. The following Figure 1 shows route of the research.

The factors that stand out in the general research results were categorized in order to establish a direct relationship with those aspects that have an impact on the contribution of students' audiovisual competences, as shown in Table 2.

**Table 2.** Codifications.

| Categories | Codes |
|---|---|
| Audiovisual competence | Self-training |
| | Innate capacities |
| | New learning environments |
| | Audiovisual production processes |
| Media literacy | Teacher training |
| | Generational digital divide |
| | Interaction processes |
| Digital multimedia tools | Video editors |
| | Image editors |
| | Social networking |
| | Interactive presentation apps |
| Sustainable educational environment | Flexible curricula |
| | New roles of educational actors |
| | Continuous innovation |

**Purpose of the research:**
To diagnose the development of audiovisual communicative competencies handled by Ecuadorian adolescents, during the Covid-19 pandemic, as a contribution to sustainability, based on the intervention of educational actors.

**Research question:**
Do the audiovisual skills acquired in a self-taught by Ecuadorian adolescents during the confinements to reduce the spread of COVID-19 constitute an integral capacity to perform in society, and do they involve changes in higher education curricula, specifically in social communication careers?

**Research methodology:** descriptive qualitative research.

**Research instruments:** 1) Focus groups, 2) Semi-structured interviews, 3) Non-participant observation.

**Results:**
- Perceptions towards the development of audiovisual competences of teachers, students, and parents.
- Experts' views.
- Non-participant observation products.

**Conclusions**

**Future lines of research**

**Figure 1.** Research outline.

The relationship between the perceptions of parents, teachers and students shows four central aspects as axes of the development of audiovisual competences, from an integral vision and in relation to education projected into the future. The self-training capacities of all educational actors, the interaction processes generated around the development of these competences and the technical, communicative and curricular challenges assumed for sustainability are taken into account.

*4.1. Teachers' Perceptions*

The teachers mention that the students have shown skills and creativity in making videos, but they also point out that the students need to improve their ability to synthesize information and reinforce academic competences. They also indicate that as tutors of the process, they have had to develop their own audiovisual competences and self-educate themselves in the production and editing processes.

On the other hand, they highlight the institutional and socio-economic limitations faced by some students in creating audiovisual content. Therefore, teachers expect universities to strengthen students' audiovisual competences with a focus on the development of critical thinking and communication skills.

Class suspensions to avoid COVID-19 contagions involved asking students for assignments through audiovisuals. "We did it at the beginning, but we realised that they were limited to downloading videos and the content was not clear to them" (P-L-9), so "the children were asked to present a topic through TikTok and creativity was awakened" (P-L-4).

The teachers' opinions are divided regarding the students' abilities to produce audiovisuals. "It has caught my attention by the fact that the young people investigate and make videos easily" (P-L-7). "They have surpassed themselves because they do impressive work. I have seen children's work, who record videos and share their experiences" (P-L-2). Students "use these resources well for their social networks. However, they lack synterisation of information. There is a lack of reinforcement of academic skills" (P-Ib-4).

Teachers pointed out that they had to learn about audiovisual language and form their own media competences before requesting audiovisual resources from their students. "It was difficult for me to integrate into the digital world. I had to learn Canva, Genially. It has helped me because the kids can also research, contribute, present their work, so we are getting them interested in the class, trying to get them to interact" (P-L-5). "This pandemic has forced both teachers and students to be at the forefront of the media, so that the classes are dynamic and we have the participation of all students" (P-L-4).

Audiovisuals are a way of motivating students to participate. "I use freely available videos from the internet to plan and show my classes and in the same way I encourage them to participate" (P-Ib-6). In classes, "we use different audiovisual resources and at the end we ask them to make an explanatory video to show the process of the finished product. We always try to make the videos in applications that they can work with" (P-Ib-5).

Teachers point out limitations related to institutional conditions and socio-economic levels of students that condition audiovisual creation. "I am a public teacher. What happens when students don't have access to these tools or don't have a connection? The challenge arises, one has to adapt to what the students have at hand" (P-Ib-10), and "being a public institution, we have mainly economic shortcomings in order to have adequate technological resources. Face with this, it had seen a strength, the creativity of the students, and this is something good that they have shown" (P-Ib-1).

In the private sphere, there are other scenarios: "they are children who have elements to work on audiovisuals. The pandemic, in a way, was not so hard because the kids already knew about types of shots, framing, so it was easy. Some of them already knew how to edit in Adobe Premiere" (P-L-3).

The teachers hope that upon entering university, students will strengthen their audiovisual competences, but "we have to differentiate the age, academic level and degree of responsibility of the children because it goes hand in hand with their academic performance, as they have deficiencies in the process of researching and presenting final products" (P-Ib-2).

From childhood "we should train, and so should the university, critical, analytical, reflective professionals. We should develop in them why and for what purpose we study a subject in our lives [...] only in this way could we change societies" (P-L-7). "There should be audiovisual literacy, because there is a saturation of information [...] there is no discernment, choosing what to see and how to interpret it [...] I think that universities should give us this training so that we can reproduce this information for our students" (P-L-3).

In secondary education, "most young people dream of being influencers, of being YouTubers, they know how to create content" (P-L-3). Based on this fact, "universities can focus on strengthening skills, we already have the creative aspect, we need to delve deeper into audiovisual technology" (P-L-6). "What the university and public policy have to promote is the generation of professionals with audiovisual communication skills that

help to develop students' critical thinking, with which we will be achieving learning for life" (P-L-10).

*4.2. Students' Perceptions*

During the focus group discussion with the students, some indicated that they acquired audiovisual editing skills before the pandemic, in self-taught learning. Participants show knowledge of the roles and processes of audiovisual production. Most indicated that they pre-plan the production of photographs or videos, although not all are familiar with the term "scripting". It is worth noting that the students refer to aspects of care for the quality of the audiovisual products they produce, as well as the importance of carrying out light and sound tests prior to editing the product. In addition, they indicate that the development of their competences is not only for academic purposes, but also for recreational ones.

Some students indicated that they know little about audiovisual production and that their skills were developed before the confinements, in the same educational centres, although they also did it via self-taught learning. "The truth is that I'm not very good at audiovisual production, because I don't use it much, I don't like filming" (E-L-2). "I don't [know] much, but in the educational institution they gave us workshops on visual arts and taught us how to take pictures, the types of framing" (E-I- 2).

One student pointed out that "I have had some studies regarding audiovisual production and also in terms of practice" (E-L-1). Another stated that "I think that in pandemic we have learnt programmes for editing images or videos, so I don't consider myself an expert, I have learnt at school itself" (E-I-1). Another revealed that "I have had to work empirically, especially on projects. I had to start to learn, go through some videos, understand how editing works and create something coherent" (E-I-3).

In another area, most of them plan to make photographs or videos. "Personally, when I want to take a photo to upload to my social networks or to promote a product, I first think about who the photo is aimed at" (E-Ib-2). Another participant indicates that "there should be prior planning because you can't make a video without planning, without knowing what you are going to do" (E-L-3). They recognize the prior preparation; however, they do not know the term script."If there should be a plan" (E-SD-8), "I think there should be a plan, because you have to see the angle so that the person does not come out wrong or if it is a school subject you have to go over it so that the video comes out well, and you do not repeat" (E-SD-7).

One student stated that "sometimes you can improvise, for example, in the TikToks nowadays most people improvise" (E-Ib-5), but "to upload to social networks you need more elaboration and a previous plan" (E-Ib-4). In general, "everyone has different planning. For example, to make a video or a photograph, we would have to see where the cameras are going to be positioned, the lighting, the sound, the environment" (E-L-5).

When asked to point out three concrete steps involved in planning photography, it was indicated "to check the light, the place and also the weather conditions because in open spaces, if the weather is not favourable, artificial lights are needed, and that the camera lenses are clean" (E-Ib-2). Lighting and angle criteria are predominant: "Choosing the space, the lighting and that the object I am going to photograph is there" (E-L-9); "The lighting, the angle and that my camera is clean" (E-L-8).

When inquiring about the roles in an audiovisual team, in the discussion groups held in Ibarra, Santo Domingo and Loja, the following roles in an audiovisual team were identified: cameraman, director of photography, director, editor, scriptwriter and producer.

In order to maintain quality standards, such as an audio recording, students indicated that they monitor environmental conditions such as "a previous voice take to know if there is any problem" (E-Ib-3). "Record in a room with no echo" (E-L-6), "go to a closed place, record a specific part" (E-L-1) or "check that there is no noise or interference and vocalization is also very important" (E-Ib-4).

The use of audiovisual technologies occurred "when I was eight or nine years old, I started filming and taking photos [...] nobody taught me, one discovers by oneself [...] at

school, I had to record videos and edit them, when I don't know something, I look for a tutorial" (E-L-3). "About six years ago. I used to watch how they did it because my mum couldn't afford to buy me a phone, and I started to learn how to use it" (E-L-2).

In addition to the use for academic purposes, there is recreation. "I use the devices for recreation" (E-L-3), "I use my mobile phone to communicate, but I also record stories for Instagram or to inform my family where I am" (E-L-5). The most frequent are combined uses, "creating content for the social network TikTok but mostly for homework" (E-L-6), "school work and content creation" (E-L-9).

In the three cities, the participants know and have used the following audiovisual editing software: Adobe Audition, Adobe Photoshop, Filmora, iMovie, Inshot, Video Start. Some of them discovered these tools out of necessity or at the suggestion of their teachers. However, their training is self-taught. Three participants, one per city, indicated that they learned to use Adobe Photoshop at school.

Other software cited is related to the demands of the tasks, as one student indicated that "before the pandemic I didn't like to record, I was shy, but I was forced to because there were tasks where I had to record videos [...], I have used Filmora to edit videos for school" (E-L-9). Another software is Inshot, with one student saying that "I have used *Inshot*. It was recommended to me at school. I learned by myself" (E-L-6). "In the pandemic I learned to use *Inshot*, because I needed to record and send projects [...] we asked the teachers for help, and they suggested it to us" (E-L-7).

Students are interested in learning expert audiovisual editing software, "the computer programmes that professionals use" (E-Ib-5). "I would like to learn other programmes for editing videos or photos such as Photoshop" (E-Ib-7). "More video and animation applications, together with all the functions they allow" (E-Ib-1) to "edit videos, change the background" (E-SD-8), "make animations" (E-SD-9), "more applications or programmes for editing" (E-Ib-7) because "I don't know anything about that" (E-SD-5).

In relation to audio processing, it was indicated that "I am interested in learning Adobe Audition, it has many mechanisms. Although I have seen videos, it has been complicated for me to learn" (E-L-1), and "I am more attracted to sound, so I am interested in learning how to use Adobe Audition" (E-L-2).

Learning the applications happened thanks to the guidance of the teachers. "The teachers were in charge of teaching us how to use the application to make our work easier" (E-SD-5). "Some of the free apps I used before because I like to take pictures. Now, with the tasks through videos or pictures, I can implement what I had learned before" (E-Ib-1).

### 4.3. Parents' Perceptions

In general, most parents consider themselves as partners in the audiovisual creation of their children, with the latter having a more advanced knowledge of the subject. They recognize their children's contribution to their own learning, and in many cases, it is the children who teach their parents. Regarding their own audiovisual skills, it is indicated that the pandemic has forced them to acquire superficial knowledge, but they still see themselves as learners compared to their children. The parents participating in the focus group present different levels of knowledge, from those with minimal skills to those with professional skills.

Parents' testimonies about their audiovisual skills range from minimal knowledge to professional use. At the first level are those who say, "not really, I don't know much about audiovisuals. It has been hard for me, now with my children, to edit videos" (Pd-L-5). "My children have helped me with technology, I don't know anything, I know the basics. When I need help, I ask them for help. I have a business where I don't need technology" (Pd-L-6).

Then, there are those who report low proficiency: "I know very little, [about] photographs, videos, I know the basics, the elementary" (Pd-SD-4); "The basics, what the phone gives" (Pd-SD-3); "Not professionally, I share what a professional does in the institution where I work" (Pd-SD-1); "Now it is easier, before we had to take a camera, take the photo

and develop it, now everything has changed and it has become much easier. But we can focus, frame something, get a good angle and colour" (Pd-Ib-5).

Others say that they have audiovisual production skills. "I consider that I do have the ability because my profession is journalism, I have worked in it for almost two decades, and it has forced me to handle photography and video well" (Pd-Ib-4). "I love photography, it is my hobby, I handle from analogue cameras to mobile phones, I have enough knowledge for the process of taking a photograph" (Pd-Ib-6). "I know how to create content for my work. Generating videos, uploading pictures or sharing. My daughter just showed me how to add effects, movements in a video, which is difficult for us" (Pd-L-3).

In the coexistence around audiovisual creation, a concept of parental accompaniment predominates. "In the pandemic, we only provided them with the computers, and they have been learning on their own" (Pd-L-1). "I didn't need to teach my son about audiovisual production, because he has taught me. I haven't had conversations about this because he always presents me with very well—done work" (Pd-Ib-2). "I try to help my daughter as much as I can in editing, but she rather teaches me" (Pd-SD-3).

Parents' testimonies highlight the differences in knowledge with their children's generation. "My son is in the third year of baccalaureate, since the pandemic he has been in charge of his homework, he has hardly ever asked me for help, I don't know if it was necessary, but what can I ask him if I have no idea" (Pd-SD-2). "It was hard for me to adapt; digital trans-formation arrived through the pandemic and forced us to transform ourselves. I was traditional, but he had new applications and was very advanced" (Pd-Ib-4).

In response to the question about parents' learning of languages and audiovisual production to help in the education of their children, the greatest contribution of adolescents stands out; parents recognize their minimal participation and even a role as learners of their children's discoveries. "Before, you took and recorded something simple on your phone. But in the pandemic, I had to learn, to look for ways to help, especially my younger son, because the older one sometimes knows more, and together we learn" (Pd-SD-5). "I have two scenarios, my 17-year-old son manages the video production, but the 6-year-old depends on me, I talk to him in simple terms. With my older son, there is a dynamic conversation to improve his production" (Pd-Ib-6). "My 18-year-old daughter uses Instagram and Facebook as a sales network and takes pictures, produces them with quality [...]. She relates content with motivational phrases to people [...]. She has no communication training" (Pd-L-2).

In the pandemic, "by necessity we both had to learn, but he won, I stayed, he handles these applications well. The pandemic forced us to prepare ourselves and keep searching. At the beginning [making] videos took us a long time, now the process has been shortened" (Pd-L-5), "he [my son] was the one who gave me feedback on what I learned when I was training" (Pd-SD-1).

Parents of adolescents would like their children to learn "video editing" (Pd-SD-4): "more video editing and be more organized in terms of content" (Pd-SD-5). "They should go deeper in the area of video, in editing, because in the pandemic we discovered their love and ability for this" (Pd-Ib-1); "about narrative, construction of the message, script, pre-production, the bases that intervene in the quality of the final product" (Pd-Ib-3); "about pre-production, lighting, techniques that they know, but empirically. Also, how to handle a professional camera because it has lenses, lenses, filters" (Pd-Ib-5).

In the discussion group, it was pointed out that "young people master a basic level of audiovisual resources, they do not handle programmes nor do they have the necessary equipment at a professional level, which allows for an application, but no more than that" (Pd-L-4); on this basis, it is understood that "if they want to move on to making a professional video, they need audiovisual training. There is the empirical level and if they do well with luck they can go far, but I am convinced that for any profession you need training" (Pd-L-2).

In addition to the forms, depth is sought: "I would like them to focus on the part of giving a good message, something that contributes to others, not just on the image"

(Pd-Ib-2); "what they handle is basic knowledge, everything is homemade" (Pd-L-5);- "they have managed to handle the situation quite well. I think we need to work on the person as such. Improve the presentation and the content so that it is of better quality" (Pd-L-3).

### 4.4. Expert Views

The results of the interviews point to the impact of the pandemic on education, highlighting the importance of integrating technological and audiovisual skills into academic training. They highlight the need to combine theory with practice, adapting content and teaching methods for distance education. It is also proposed that audiovisual skills should be a core subject in education, regardless of the specialization being studied.

The responses discuss the need to restructure the profile of the communicator in order to integrate the changes derived from the innovations of the profession and technology. On the other hand, the importance of teaching the responsibility of the journalists and their impact on society is reiterated. In summary, it is suggested that the pandemic has accelerated the coupling of education to new technologies and trends.

The suspension of classes due to mobility restrictions during the pandemic led children and young people to "accelerate the development of technological competences" (Interviewee-2) so that they could participate in a "remote modality of improved learning, with respect to didactic resources, but which is disordered knowledge with a lack of background" (Interviewee-1).

In this environment, educational institutions play a relevant role by integrating processes and tools. On the one hand, it is recognized that "the trend towards virtuality is undeniable, we cannot turn back, we must start thinking about new paradigms that allow us to adapt methodologies and open up new perspectives" (Interviewee-10), and on the other hand, "education must combine theory with empirical experience. The University must value both within its curriculum" (Interviewee-1), "it must be transformed; it has always focused on the theoretical, purely on the academic part, it is important to learn by doing" (Interviewee-6).

Then, virtual spaces were created to continue with the classes, despite acknowledging the physical and skills limitations of the learners, but fortunately, some young people already had knowledge. What possibly did not exist was a continuous habit and a technique that had to be deployed for virtual classes and content creation, intensively and periodically" (Interviewee-4), which helped them to have "technological competences [...]. This has been fuelled by the pandemic issue, not only for an academic reason, but also for a leisure reason" (Interviewee-3). "In relation to previous generations, they already have a basic knowledge of audiovisual language, of audiovisual narrative, even, because they consume a lot of series, clips, content on social networks, and they also produce it" (Interviewee-4).

During the days of isolation, many young people made "explorations of the possibilities of technologies" (Interviewee-8) "from their context, from their use" (Interviewee-10); to the relationship between theory and practice is added the virtual environment, from which they tried to continue teaching in dynamics close to the classroom experience, but "they have not adapted the contents, the ways of teaching to make up in part for the absence of face-to-face education" (Interviewee-8).

It seems that the handling of technologies forms the competences that conventional education provides for students, but competence "must be seen as an aim, not as a means [...] it is a construct that combines skills with knowledge and aptitudes, therefore, [...] it is important not to confuse the ease with which self-taught people manipulate technologies with competence" (Interviewee-7).

Thus, a scenario was configured in which the audiovisual competences of adolescents and young people were enhanced because they had to interact through online applications and show their learning in audiovisual productions, videos and photographs. The relevance of changing training paths, plans and offers in order to take advantage of self-taught competences is questioned. One interviewee pointed out that media and audiovisual skills "should be a core subject regardless of the specialization being studied" (Interviewee-1).

In the case of communication careers, there are frequent updates due to the commitment to informing people and technological innovations. Communication "is not a career that was left behind, it goes pace with the galloping digitalization, with the technology that has created this need to send clearer messages to society" (Interviewee-5), faced with the urgency of "updating the curricula, we realized that there are years of delay in the way the market moves, the pandemic brought us up to date" (Interviewee-4).

First, "we must differentiate between a technical career and a career in communication, in communication we learn how to create good content, a message, how to reach the audience" (Interviewee-6). "What defines the value of a communicator is their ability to communicate, to have something to say, a content, their ability to analyse a certain reality and generate an opinion on that reality" (Interviewee-8). The applicants "must take into account what they want to be, a technical expert, a technology developer or someone who understands what should be sought and what the intention of the communication is" (Interviewee-9).

Then, "the academic units must take into account the scope of their vision of training, what they want to teach and who they want to train" (Interviewee-9), "we must understand that communication is not an instrument, it is a process" (Interviewee-2). "It is important to review and make prospective attempts of where we are and where to project our curricula" (Interviewee-8).

It is also necessary to "restructure the profile of the communicator. We have to make changes in relation to traditional and classical teaching because we have an interdisciplinary perspective" (Interviewee-4). Today, "the attraction for a communication student is no longer to be the best presenter, but to receive tools to enhance an image in social media, we are in times when the ego comes first and the collective has moved to the third plane" (Interviewee-5).

Regarding the students, "when they approach the communication degree course, they see it as technical, especially audiovisual, that they are going to learn tools, but they come to understand that it is not enough, what is important is the management of language, narrative, and strategy" (Interviewee-4), hence the relevance of explaining the possibilities of audiovisual narratives.

In "all communication subjects, although more so in the audiovisual field, an update is necessary because they have been affected by the rapid cycles of technological innovation and the uses, demands and customs that have resulted from the pandemic" (Interviewee-8). The updating of training programmes in communication "should be oriented towards the use of tools that allow for better information management [...] without neglecting the need to update the curriculum, they should improve practices, the use of technology, starting from the basics" (Interviewee-8).

> "Thanks to the series, the impulse of Netflix and other platforms, the language of television and film embrace each other, merge, create a hybrid language [...] it is necessary to break down the frontiers that separate cinema and television to enter the audiovisual language of series. We must take advantage of the richness of this format to strengthen our interpretative and narrative skills, and learn to understand and appreciate the complex plots and characters that this mean offers us [...]. A communicator needs to learn how to make a three-minute content, which has the quality required to have an impact on another person, how to handle YouTube". (Interview-do-5)

> "If a student learns film, television in a communications degree, he can transfer it to the digital world. When you are going to tell a story you must have a basis in journalism, it's no good if you know how to handle Facebook, if you don't know how to structure a story [...] you can know how to use tools, editing programs, but that doesn't guarantee that you are going to make good content". (Interviewee-6)

Today, knowledge is acquired through audiovisual transmission, thanks to the democratization of information" (Interviewee-5). Universities "must have the capacity to train competi-

tive students, whose skills can demonstrate that it is worth hiring them" (Interviewee-8), but what they teach is a "tacit and disorderly knowledge that seeks to satisfy audiovisual skills from immediacy. Therefore, we must start with theorization for its empirical prospection" (Interviewee-1). The university "must strengthen the content, when the content is good it will be watched, even if the audio and image quality are not the best" (Interviewee-6), and it is also requested "to prepare young people to use the medium, the audiovisual language to generate their own businesses and provide advice related to digital" (Interviewee-5).

Education is in constant renewal, and although a rapid updating of competencies has been demanded during the pandemic confinements and there is speculation about changes in the training programs in careers related to communication sciences, a global questioning of the education model is not left aside; thus, it is understood that despite the fact that "it is stated that most universities use a constructivist model, however, we continue to maintain the conventional practice. The paradigm of the university professor should be essentially reformulated, changing the model of how the student is learning" (Interviewee-10).

It gets to the point of proposing that "it is the teacher who should change and not the audiovisual content" (Interviewee-5), and there are teachers "who dominate the theory and referential authors, but in practice they do not know how to apply it" (Interviewee-6). Universities should "have very up-to-date teachers and a direct connection with reality, which allows them to create resources and be useful in the social environment" (Interview-do-8).

Teachers "must get involved in this new knowledge and if they don't know it, they must learn it, taking advantage of the fact that the tools are very intuitive" (Interviewee-6). "I don't think we should reduce ourselves to the technological or instrumental field, the challenge as teachers and researchers is to go beyond that and work on the content" (Interviewee-2). "The challenge is how to ensure that teachers educate from a critical, comprehensive and socio-cultural view of the media, not just a technological view" (Interviewee-7).

Regarding the way in which work will be carried out once COVID-19 is under control, it was indicated that "distance or semi face-to-face education as modalities of attention has many possibilities and even more so in combination with technological tools" (Interviewee-2), "several experts have warned that in post-pandemic scenarios the use of tools such as Zoom, Meet or others will be permanent, but there are difficulties such as the digital divide, and the need for physical contact" (Interviewee-3), but it is important to distinguish whether only virtual environments are required as inputs, since "if the concern is to use a tool, a university education is not necessary" (Interviewee-9).

Face-to-face training "is very difficult for us to give it up, because human beings are sociable, they seek to learn with others and that does not mean that with respect to virtual platforms they do not learn, however, that closeness is very important" (Interviewee-10). "The training process requires contact, looks, a physical space for discussion, that does not mean that the virtual is inferior, they are complementary" (Interviewee-7); it is not a question of undervaluing virtuality, "I do not discard virtuality. There are things that can be done virtually and others will have to be done face-to-face" (Interviewee-9). It was recalled that "in a truly virtual education, the greatest percentage of responsibility for the process as such goes to the contents, so that they are attractive" (Interviewee-8).

The challenge of the "academia is to agree the humanities and technologies together. Technical skills are needed, but they must be channelled into the humanities" (Interviewee-3). Universities "should ask themselves the question of the relevance of the media and communication channels, in many things it will not be possible to resort to technical resources" (Interviewee-9), "I believe that the role of the academy is to match theory and practice, to think about these integral communication processes, so that they become transformative" (Interviewee-2).

In the university, "the responsibility of the journalist must be strengthened" (Interviewee-6), and "we have a strong impact through our messages, so what impact are we generating? This is a plus for the university, because it can help to build a better society or at least question when it is not working well" (Interviewee-4).

### 4.5. Results of Non-Participant Observation

After the focus groups and expert interviews, non-participant observation was applied with 10 students in general basic education and high school through a workshop that lasted five days. This process of enquiry consisted of three moments. The first was to carry out a diagnosis of the audiovisual competences and digital multimedia tools acquired empirically. The second was due to the training given to the students on technical elements of production. Additionally, the third served to evaluate the skills acquired as a result of the application of the practical workshops as a result of a video.

The audiovisual productions made by the adolescents during the non-participant observation are hosted in the open repository Zenodo https://bit.ly/3fpLQsd https://bit.ly/3xWV0Db, (accessed on 20 January 2023) [73]. The evaluations given to the students' work, before and after the audiovisual productions, are shown in Table 3.

**Table 3.** Results of non-participant observation.

| **Diagnosis, Prior to Audiovisual Production** | | | | | | |
|---|---|---|---|---|---|---|
| **Criteria** | **Evaluations** | | | | | **Total** |
| | **1** | **2** | **3** | **4** | **5** | |
| Distinguishes angles and types of image shots | | 2 | 4 | 1 | 3 | 10 |
| Recognises the audiovisual process | 1 | 2 | 2 | 3 | 2 | 10 |
| Understands the roles of audiovisual production | 2 | | 4 | 2 | 2 | 10 |
| Internalises and shows understanding of the stages of audiovisual production | 1 | 3 | 1 | 4 | 1 | 10 |
| Displays skills in handling audiovisual equipment | 1 | 1 | 3 | 2 | 3 | 10 |
| Capture quality images with their devices | 1 | 1 | 3 | 4 | 1 | 10 |
| Participate in the creation of the script | 1 | 1 | 1 | 1 | 6 | 10 |
| Demonstrates knowledge of audiovisual production software | 1 | | 3 | 4 | 2 | 10 |
| Know about the virilization of the audiovisual through social networks | 2 | 3 | 4 | | 1 | 10 |
| Total | 10 | 13 | 25 | 21 | 21 | 90 |
| **Evaluation of the finished product/video** | | | | | | |
| **Criteria** | **Evaluations** | | | | | **Total** |
| | **1** | **2** | **3** | **4** | **5** | |
| There is coherence between what was planned and what was achieved | | | 5 | | 5 | 10 |
| The concept that the student and the team wish to convey is understood | | | 4 | 5 | 1 | 10 |
| The images and sound are of an acceptable quality, which implies acceptance by the audience | | | | 9 | 1 | 10 |
| The composition of the image is adequate, there is a logic to the composition of the image | | | 5 | 5 | | 10 |
| The student's contribution to the final result is appreciated | | 1 | 3 | 3 | 3 | 10 |
| The time and quality of the audiovisual product achieved supports other versions for social networks | | | 9 | 1 | | 10 |
| In free space dialogues and while working in a team, he/she mentions or alludes to social networks, *YouTube, Facebook* or other applications | | 4 | 4 | 2 | | 10 |
| The participant together with his team achieved the set goal | | | 4 | 1 | 5 | 10 |
| The participant follows a rhythm or routine of using devices to capture video or photographs | | | 6 | 3 | 1 | 10 |
| Total | 0 | 5 | 40 | 29 | 16 | 90 |
| **Instrumental skills** | **Evaluations** | | | | | **Total** |
| | **1** | **2** | **3** | **4** | **5** | |
| Time management | | | 1 | 4 | 3 | 2 | 10 |
| Problem-solving | 1 | 1 | 1 | 4 | 3 | 10 |
| Decision-making in a real situation | 1 | | 3 | 2 | 4 | 10 |
| Planning | | | 1 | 6 | 3 | 10 |
| Use of computers and/or computer/mobile phone applications | | 2 | 4 | 4 | | 10 |
| Database management (information search) | | 3 | 3 | 3 | 1 | 10 |
| Development of oral communication skills | 1 | 1 | 2 | 3 | 3 | 10 |
| Development of written communication skills | | 1 | 2 | 5 | 2 | 10 |
| Self-initiated research in relation to the proposed work | | 1 | 5 | 1 | 3 | 10 |
| Total | 3 | 10 | 25 | 31 | 21 | 90 |

**Table 3.** *Cont.*

| Interpersonal skills | Evaluations | | | | | Total |
|---|---|---|---|---|---|---|
| | 1 | 2 | 3 | 4 | 5 | |
| Self-motivation | 1 | 1 | 2 | 3 | 3 | 10 |
| Ethical sense | | | | 8 | 2 | 10 |
| Interpersonal communication | 1 | | 3 | 3 | 3 | 10 |
| Group work | 1 | 1 | 1 | 5 | 2 | 10 |
| Conflict management | | 1 | 3 | 4 | 2 | 10 |
| Negotiation | 1 | 1 | 4 | 4 | | 10 |
| Leadership | 1 | 2 | 2 | 2 | 3 | 10 |
| Total | 5 | 6 | 15 | 29 | 15 | 70 |

The following comments are reproduced from the researchers' anecdotal record:

- Students contribute ideas to the group when problems arise. They have theoretical knowledge, but there is a lack of practice and handling of the instruments.
- Team organization and active participation during the process is appreciated.
- There are students who master editing applications and basic audiovisual language, but others do not.
- Most of the teenagers showed interest in the planning phase; however, when the filming started, all of them got involved.

Based on the premise that through non-participant observation, the researcher does not intervene in the life of the observed group, but instead interacts, it was identified that only half of the students distinguish the angles and types of image shots, understand the roles of audiovisual production and are familiar with the issue of virilization of audiovisuals through social networks. It is imperative to highlight that they all recognize that they have participated in the creation of scripts.

In terms of participation in the training workshop, the students contributed ideas to the group when problems arose. It was evident that they have theoretical knowledge, but there is a lack of practice in handling equipment. They showed active participation during the process, and there are even students who mastered basic audiovisual language and editing applications.

In the last stage, the finished video was evaluated, in which the quality of the editing of images and sound stands out; on the other hand, the time taken for the audiovisual narrative facilitates convergence towards other platforms and applications, especially the leap to social networks, and finally, it is notable that the students follow the routine in the use of devices to capture photographs and videos, contributing to the development of skills in audiovisual production. In relation to the students' technical competences, progress in planning and research stands out as a starting point; in the same way, skills in terms of written communication predominate, taking self-taught training and innate abilities as a basis. Similarly, with regard to interpersonal competences, they show a positive adaptation to group work and a high sense of ethics.

## 5. Discussion

The use of audiovisuals to support the teaching–learning process was confirmed in the GDD. The teachers pointed out that the students applied their skills to elaborate audiovisual contents; in many cases, they showed high competences regarding the conception and construction of audiovisual pieces, although absences of argument competences or integral conceptions were identified, in addition to material restrictions linked to the students' socio-economic levels.

On the other hand, both teachers and parents converge in pointing out the important role of the university in educating qualified professionals, with audiovisual competences, so that they can contribute to equitable societies that imply human and sustainable progress. In view of the intention of teenagers to create content for social networks, the relevance of generating value and critical thinking is reiterated.

Despite rapid technological advances and popular trends, higher education often adopts a conservative and traditional perspective, which can call into question the relevance and appropriateness of university degree content to societal and business demands. It is necessary to question whether academic institutions are adequately embracing and adapting new technologies and trends to ensure that students are prepared to meet the challenges of the evolving world.

In the GDD, it is also evident that adolescents show different levels of audiovisual competences, acquired in self-taught learning, but also in the educational centres, before the beginning of the pandemic, they indicated that they received accompaniment from teachers and parents, and they are interested in learning more software to build stories that allow them to participate in the society mediatized by social networks on the internet.

Audiovisual competences are enhanced through practical and self-taught learning, which even seem to come from the students' innate abilities, where, although the lack of reflective knowledge and theoretical support is visible, the students present clear ideas about audiovisual production processes. Similarly, the study shows the importance of media literacy, making visible the generational digital gap between teachers and students, the importance of teachers developing audiovisual competences, as well as the need for interaction processes that strengthen the skills developed and turn competences into social contributions.

The results also show a wide use of multimedia digital tools: editors, programmes, applications and social networks, through which experiential learning is developed. In this way, the need to generate a sustainable educational environment is relevant, in which it is essential for curricula to be flexible and adaptable to the needs and challenges that arise in relation to sustainability.

An important distinction that experts point out is between instrumental skills and the knowledge necessary for effective communication. Although adolescents may have skills in editing tools and appreciation of audiovisual languages, this does not guarantee that they possess media competence acquired through formal communication studies. A combination of skills and knowledge in areas such as culture, history and deontology is required to achieve the comprehensive competence expected in graduate profiles.

One of the difficulties in current education is the gap between the teaching environments known to teachers and the virtual multidimensionality in which students develop. To address this situation, a convergence between the technological and the socio-cultural is essential, which would facilitate a more complete training which is adapted to the demands of today's world.

Specifically, the consolidation of environments that help equal opportunities for education is expected, as referred to in [74] (p. 17).

> Media education in Latin America must be understood politically from the perspective of equality: the most unequal region on the planet must integrate technologies and their use and critical training therein in the classroom, from a perspective that gives way to conditions of equity or possible futures.

> In order to achieve the above, it should be remembered that

> Preparing young people to critically understand the techno-social environment and the phenomena associated with it, and to successfully confront the impact of the media on all dimensions of life, is today a fundamental component of the right to education and an inexcusable duty of education systems [19] (p. 55).

Practically all the experts interviewed agree that the new professional opportunities in the audiovisual sector are directly linked to the digital scenario; they state that production and distribution technologies facilitate the multiplication of content and that these are designed with the viewer's greater involvement in mind.

Previous research on higher education in communication has defined new professional profiles that have emerged after the consolidation of the digital environment and that coincide with the trends resulting from this research. In the immediate future, specific roles

will be in demand: designers, programmers and developers of applications and software; specialists in digital marketing and new audiovisual platforms; creators of virtual scenarios and 3D designers [27].

> The evaluations and behaviours of adolescents do not differ from the actions of young university students, as is the case when contrasting the development of communicative and audiovisual competences of students in the first semesters of the social communication and journalism programme at the University of Tolima. Both the adolescents observed and the young Colombians achieve similar skills.

> The participants were able to recognize the importance of planning an audiovisual product from its different stages (pre-production, production and audiovisual post-production), as well as from teamwork and the designation of responsibilities to elaborate the final products of the workshop, in both cases, video clips. The students understood the different technical and compositional aspects of an image, the audiovisual language and the incidence of lighting to obtain videos and/or photographs with better technical conditions [75] (p. 79).

## 6. Conclusions

In the field of communication studies and related disciplines, there is a gap between the autonomous training that young people receive in audiovisual techniques, processes and languages before entering university, and the academic content currently taught. This prior instruction is acquired without the social context, responsibilities and ethics necessary for comprehensive media literacy. In addition, social networks and online learning, driven by the pandemic, have fostered the development of skills and attitudes that do not necessarily correspond to a complete media literacy education.

Therefore, the role of the digital teacher becomes important for the integration of the four pillars of education (knowing how to know, knowing how to do, knowing how to be, knowing how to live together) within the pedagogical process and in convergence with the demands of the information and knowledge society.

Higher education faces the challenge of adapting to the emerging interests and demands of young people, especially in terms of the role played by Information and Communication Technologies (ICT), the need for a more international education and the increase in relationships mediated by virtuality, in a context marked by the COVID-19 pandemic. Higher education is expected to emphasize the promotion of the human being and contribute to the consolidation of a more empathetic and resilient citizenship in the face of the changes and challenges of the currient world. It is necessary to reinforce students' audiovisual competences, as Ferrés and Piscitelli [16] express, so that they assume greater social commitment to their environment.

It is interesting to observe some of the new competences that teachers and researchers in the field of communication point out as essential, from an academic point of view, for the exercise of the profession. For example, Gil and de Miguel [76] claim the importance of education in values in the face of the predominance of purely technical competences. Marfil [77] also emphasizes the need to develop critical capacity as an essential element for training good professionals, while Sánchez and Peña [78] note the need to introduce personal and social competences into curricula to ensure correct personal development in a highly competitive professional environment.

On the other hand, other contributions underline the need to introduce media literacy more forcefully into the curricula of communication degrees, both transversally and by programming specific subjects on the subject, in line with the recommendations of international organizations [79]. In this way, curricula would be at the forefront of technological changes based on deontological principles and in favour of values that build democratic, tolerant and responsible societies with regard to the sustainability of nature and people.

The research carried out by Ferrés [17] (p. 107), entitled "La competencia en comunicación audiovisual: dimensiones e indicadores" (Competence in audiovisual communication: dimensions and indicators), conducts a critical analysis (assessed by more than

50 experts in audiovisual communication in Ibero-America) to define the future of education in audiovisual communication and mentions how communication professionals focused on this area should have indicators that facilitate the development of projects and their evaluation, in the face of the processes of orientation and intervention using audiovisual production.

Therefore, it is required that the dimensions and indicators of media competence described by Ferrés [14] are articulated transversally in the different areas of knowledge and, in turn, in the different components of the curriculum: objectives, contents, methodology, activities and assessment systems in the teaching and learning process in the non-university stages of education [80] (p. 138).

The need for universities to establish adequate conditions to promote student-centred learning, using innovative teaching methods, forming critical and active citizens, willing to put their knowledge at the service of society, should be highlighted [81,82], thus forcing educational actors to assume new roles within pedagogical practice, promoting critical capacity and problem solving from a sustainable vision of education.

Universities must adapt the educational processes, attending, among other aspects, to the current characteristics and needs of students, facilitating the incorporation of flexible scenarios, and where they are aware of their own educational process in the acquisition of competences and skills [83], "it is essential to adopt a global approach [...] that integrates educational technology and media education in both face-to-face and virtual and hybrid environments" [84] (p. 23).

The educational process is carried out in an integral way, involving all actors, namely, students, teachers and family, so that the development of competences is carried out to provide the necessary strength for sustainable changes and transformation towards a society that is more critical and aware of the content it produces, manages and shares.

The answer to the research question is nuanced. On the one hand, on the basis of the evidence pointed out, it is accepted that adolescents have acquired or improved audiovisual competences during the confinements that occurred due to COVID-19, but these competences alone do not constitute an integral capacity to perform in society.

In another environment, according to the experts consulted, it is pertinent to bring about innovations and updates in higher education curricula, specifically in social communication degrees. The intention is to keep up with technological changes, based on deontological principles and in favour of values that build democratic, tolerant and responsible societies with respect to the sustainability of nature and people.

Educational actors must assume new roles in order not only to transmit knowledge, but also to foster critical thinking and problem solving related to sustainability.

For future research stemming from this work, we propose to conduct focus group dialogues with adolescents and their families, as well as participant and non-participant observations of youth interactions, to determine the effectiveness of autonomous learning in emergent communication environments.

Likewise, in the future, methodological triangulations could be carried out, including quantitative instruments, to obtain responses from the educational community in the Andean area due to the proximity and shared cultural characteristics, as mentioned in the background. By having environments where the COVID-19 counterparts are better controlled, the limitations of the absence of face-to-face contacts are overcome. In this research, interviews were conducted through virtual tools that do not allow us to see the contexts or complementary messages of a direct interview, as is the case with other instruments.

**Author Contributions:** All authors contributed equally and proportionally to the conceptualization, methodological design, data collection, data analysis and interpretation, writing, revising and editing. All authors have read and agreed to the published version of the manuscript.

**Funding:** This research received no external funding.

**Institutional Review Board Statement:** The study was conducted in accordance with the Declaration of Helsinki, and approved by the Ethics Committee for Research on Human Beings of the Univer-

sidad Técnica Particular de Loja (CEISH-UTPL), dated 16 September 2021. Prior to conducting the discussion groups with adolescents, part of the methodology of the project "Horizonte humanístico en las competencias comunicativas audiovisuales y nuevas narrativas de los adolescentes ecuatorianos", we requested, through letters, the authorization of the parents.

**Informed Consent Statement:** Informed consent was obtained from all subjects involved in the study.

**Data Availability Statement:** Not applicable.

**Conflicts of Interest:** The article arises from the project "Humanistic horizon in the audiovisual communicative skills and new narratives of Ecuadorian adolescents" PROY_INV_CCCOM_2021_3156, sponsored by the PUCE Universities—Ibarra and Santo Domingo—and Universidad Técnica Particular de Loja (Ecuador), between 2021 and 2022. The purpose of the project was to identify the audiovisual communicative skills and the expectations for their academic training at the higher education level of Ecuadorian teenagers between 12 and 18 years of age, through the application of action research, for an innovative educommunication proposal from the knowledge of being.

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
