# Peer review of "Audiovisual Competences in Times of COVID-19: The Role of Educational Actors in Media and Digital Learning of Adolescents"

_sustainability, doi:10.3390/su15076323_

Round 1

Reviewer 1 Report

The ms addresses the elements that affect the audiovisual skills acquired by teenagers during the Covid-19 pandemic. Unfortunately, the ms includes shortcomings that suggest to me that it should be considerably revised before publication. I hope my comments below will help the author(s) to continue their work.

I think one of the main problems in the ms is that its main aim is not clearly stated. For the reader it is difficult to grasp why the topic is important, and more importantly, how it is related to sustainability. I suppose there could be a link, but it should be elaborated and justified.

The introduction addresses the different viewpoints to the topic, but it does not form a coherent whole where the different viewpoints are clearly linked to each other. Moreover, it is not obvious to the reader how the presented viewpoints are linked to the aim of the study.

The objective of the study is presented on page 6. How is this derived based on the previous research? Are there more specific research questions?

It is stated that as research methods focus groups, semi-structured interviews and non-participant observations were used. Please, elaborate the methods, why and how they were used. For instance, why it was important to use focus group interviews and not individual interviews? Why non-participant observations were important to carry out? How do they helped to achieve an answer to the research questions, if there were any. How were the respondents contacted and informed about the study?

In addition, consider presenting the participants in a more concise way. For instance, in Table 1 you could just report percentages of females and males, percentages of regimes, and mean ages from the different cities. Very detailed reporting might jeopardize the privacy of the respondents and make them identifiable.

Please, describe how the data was analysed. Concerning the results, I recommend summarizing them more. Now the result section is ample, but it is not obvious to the reader how the data was classified. Please, mark the quotations clearly, now some of them have quotation marks, but some do not, or only one.

Please, present the main results at the beginning of the discussion and then address them systematically linking them to previous research. In the conclusions, highlight how the results are linked to sustainability.

Reviewer 2 Report

This is a study on the audio-visual skills of teenagers. I think the significance of this research is very important, and the authors have chosen a good topic, but there are many problems in the full text that need to be revised.

The first part is the abstract, which suggests that the authors rewrite it in the form of "background-method-result-conclusion". The present presentation form confuses readers. Although I am also doing research in this field, it is difficult for me to understand what the authors have done.

Secondly, I suggest that the authors split the introduction, and the new introduction should only include the research background, reasons and contributions. The remaining part is explained as a literature review, which shows the shortcomings of the existing research and highlights the value of this study.

Thirdly, some of the results are explained in detail by the authors, but I suggest adding a diagram, including the mechanism path between all factors, so as to help readers get the results more quickly. Although the existing explanation is very detailed, it actually lacks the relationship between core factors and factors.

Finally, it is about the discussion and conclusion. The authors need to further explain the conclusion of this study and the possible reasons and shortcomings.

It should be pointed out that there are many puzzling places in the full text. For example, "[...]" in line 32,130,750, and there are still many errors in the format of references. Of course, the editing of the table needs further adjustment, and the contents of the table are very confusing now. Authors should not make mistakes in these places, which is disrespectful to reviewers and their own research.

All in all, this is a good study, but it needs further scientific revision and further adjustment according to the requirements of the magazine.

Reviewer 3 Report

I would like to thank the Authors for the opportunity to review the article submitted to the journal.

In my opinion, there are some issues that need further clarification prior to considering the paper for publication. Please refer to my specific comments below:
1) I think that the abstract is not between 250-300 words.
2) The hypotheses should be formulated.
3) It is missing here to expose the practical conclusions of the results. What do we conclude as professionals regarding the results? What does the study provide that we should take into account?
4) It would be recommended to expose main limitations or future lines to investigate in the additional section.
5) There is no in-depth literature review. The references are mostly from the old days. References could be up to date. I recommend to include recent articles from 2022 on this topic.
6) What is more, the generalizations should be avoided. Unfortunately, the generalizations appears especially in the discussion section.

I am pleased to wait for your revised version then.

Best regards. 

Round 2

Reviewer 1 Report

Thank you for revising the manuscript. I think it is now clearer and more coherent than the earlier version. There are some points to consider in order to still improve the quality of the ms.

Throughout the ms, there are lot of quotations from the references, please consider presenting the ideas more in your own words.

 It could be clarified why the experts were interviewed individually and not in focus groups.

 It was not obvious to the reader how the results from the focus group interviews represent the dynamics of the focus group discussions. I recommend presenting the focus group interview results more as a group level, not so much relying on individual's statements.

 The chapter “Expert views” is very long. Would it be possible to present the results in a more analytical and concise way, for instance, according to the interview questions?

The quotation marks are missing from some quotations in the result section. Please, add them.

Consider to narrow down the conclusions and focus on only the main points.

 You could add some reference on page 21, line 914 about the link of critical thinking, problem solving and sustainability.

Author Response

Thank you again for the detailed explanations and recommendations provided by the evaluator.

We have taken on board all suggestions.

Below are the details of the changes.

We are attentive to any new proposals that the reviewer may wish to make.

Point 1. Throughout the manuscript, there are many quotations from the references, please consider presenting the ideas more in your own words.

Response 1. The suggestion is welcomed, several ideas from other authors are written from the understanding of the researchers, but respect is maintained to indicate the origin of the quote, from the third author.

Point 2. It could be clarified why the experts were interviewed individually and not in focus groups.

Response 2. It is explained in point 3.2.2 mainly because the interviewees come from different countries and because the interviews are conduct at different times, due to their work situations.

Point 3. I recommend presenting the results of the focus group interviews more at the group level, without relying so much on individual statements.

Response 3. The researchers have placed a short presentation summarizing the main findings at the beginning of each section of point "4. Results".

Point 4. "Experts' opinions" is very long. Would it be possible to present the results in a more analytical and concise way, e.g. according to the interview questions?

Response 4. The researchers, as in the previous point, place at the beginning of each section of point "4. Results" a short presentation summarizing the main findings. We hope to meet the evaluator's expectation.

Point 5. Quotation marks are missing in some quotes in the results section. Please add them.

Response 5. The text was reviewed, and indeed, missing inverted commas were located, so the missing inverted commas have been added.

Point 6. Consider shortening the conclusions and focusing only on the main points.

Response 6. The conclusions were checked, to emphasize the relevant points.

Point 7. You could add some reference on page 21, line 914, on the link between critical thinking, problem-solving and sustainability.

Response 7. The requested bibliographic reference is included.

Reviewer 2 Report

Literature review should not be 1.1, but 2. Subsequent chapters are reordered according to this serial number.

3.5. Results of non-participant observation

As I said before, I don't think the author did a good job in this part of the analysis, but simply presented the results. It is difficult for me to directly find the laws and reasons, and this part needs to be strengthened.

Author Response

Thank you again for the detailed explanations and recommendations provided by the evaluator.

We have taken on board all suggestions.

Below are the details of the changes.

We are attentive to any new proposals that the reviewer may wish to make.

Point 1. The literature review should not be 1.1, but 2. Subsequent chapters are reordered according to this serial number.

Response 1. It is listed according to the reviewer's proposal.

Point 2. Results. I do not think that the author has done a good job in this part of the analysis, but has merely presented the results. I find it difficult to find the laws and reasons directly, and this part needs to be strengthened.

Response 2. The researchers have placed at the beginning of each section under "4. Results" a short presentation summarizing the main findings. We hope to meet the expectation of the evaluator.
